# Development and Validation of a Graded Motor Imagery Intervention for Phantom Limb Pain in Patients with Amputations (GraMI Protocol): A Delphi Study

**DOI:** 10.3390/ijerph182212240

**Published:** 2021-11-22

**Authors:** Sandra Rierola-Fochs, Luz Adriana Varela-Vásquez, Jose Antonio Merchán-Baeza, Eduard Minobes-Molina

**Affiliations:** Research Group on Methodology, Methods, Models and Outcomes of Health and Social Sciences (M3O), Faculty of Health Sciences and Welfare, University of Vic-Central University of Catalonia (Uvic-UCC), C. Sagrada Familia, 7, 08500 Vic, Spain; sandra.rierola@uvic.cat (S.R.-F.); luzadriana.varela@uvic.cat (L.A.V.-V.); eduard.minobes@uvic.cat (E.M.-M.)

**Keywords:** phantom limb, pain, amputee, graded motor imagery, treatment

## Abstract

Background: Phantom limb pain can be defined as discomfort or pain in a missing part of the limb. The aims of this study were to develop and validate, through a Delphi methodology, a graded motor imagery protocol in order to reduce phantom limb pain. Method: Physiotherapists and/or occupational therapists with experience in research and a minimum clinical experience of five years in the field of neurorehabilitation and/or pain were recruited by part of a group of experts to assess the intervention. The study was conducted through an online questionnaire, where experts assessed each aspect of the intervention through a Likert scale. As many rounds as necessary were carried out until consensus was reached among experts. Results: A total of two rounds were required to fully validate the intervention. During the second round, the relative interquartile range of all aspects to be assessed was less than 15%, thus showing a consensus among experts and with good concordance (Kappa index of 0.76). Conclusion: Experts validated a graded motor imagery intervention of phantom limb pain in patients with amputations (GraMi protocol). This intervention can help to homogenize the use of graded motor imagery in future studies and in clinical practice.

## 1. Introduction

Phantom limb pain (PLP) can be defined as discomfort or pain in a missing part of the limb [1]. It affects between 60% and 80% of people who have suffered an amputation but not with the same frequency and intensity [2]. There are different factors that influence the perception of pain, such as preamputation pain, location, sex, and time since the amputation, among others [3]. In 72% of cases, it appears in the week following the amputation [4]. However, in some cases, it can appear gradually over weeks, last over time, and become a chronic pain, causing a decrease in quality of life [2].

PLP is identified as a type of neuropathic pain and, despite its presence in the literature since 1551, the exact pathophysiological mechanisms by which it occurs are not yet known [4]. The mechanisms involved appear to be sensitization of the central nervous system, injury to the peripheral nerve, and cortical reorganization [2].

Since the late 20th century, the dominant theory about the etiology of PLP is the maladaptive plasticity that occurs in the brain after suffering an amputation [5]. On the one hand, adjacent cortical areas can occupy areas corresponding to the amputated limb, causing activation of the affected areas by stimuli from healthy areas [6]. On the other hand, after an amputation, the representation of this part of the body at the central level may remain intact, but there may be a mismatch between the visual feedback it receives and the perception of that limb [7]. Based on these two theories, an incongruity of information can occur between the order that the brain wants to give and the visual and sensitive feedback it receives in response to the pain [8].

Currently, there are different lines of treatment to decrease PLP: pharmacological, surgical to prevent or treat the formation of neuromas in the stump, and physiotherapeutic, which includes mirror therapy, sensory discrimination training, virtual reality training, and graded motor imagery (GMI), among others [9,10].

GMI is a form of comprehensive rehabilitation, developed in the early 2000s, which aims to reorganize the maladaptive plasticity that must occur after amputation [9]. It is a form of progressive treatment, designed to train and reorganize the brain based on three progressive and consecutive stages: laterality recognition, motor imagery, and mirror therapy [11]. Laterality recognition consists of differentiating, through images, the right and left side of the lower or upper extremities depending on the level of amputation, which are presented in a mobile application [11]. The degree of recognition depends on the body and brain scheme present in the person and is of great importance in the planning and reorganization of voluntary movement [12,13]. The motor imagery consists of imagining functional movements or actions without performing voluntary contraction; it is a mental simulation [9,12,14]. These first two stages have the purpose of reorganizing the representation and the cerebral scheme based on neuroplasticity, activating the premotor and supplementary areas [12,15]. The third stage is mirror therapy, which consists of placing a mirror in the sagittal plane between the limbs, so that the amputated limb is behind the mirror, and the person is asked to observe the reflection of the healthy limb in the mirror [7,11,16]. Through observation, a technique based on mirror neurons, it aims to teach the brain that there can be pain-free movement and to pre-activate injured areas to decrease maladaptive plasticity produced after amputation [16,17].

GMI is based on different neuroscientific foundations, such as neuroplasticity and the use of mirror neurons, with the idea that reorganization of the cerebral cortex could help decrease pain [10,12,18,19]. There are studies that use some of these techniques in isolation and show their effectiveness in reducing PLP [20,21,22] but few that combine the three techniques [14,19]. Even so, the evidence that exists shows a positive influence of GMI on PLP, but there is no detailed and homogeneous intervention [14,19]. Therefore, the aim of the present study was to design and validate, by a committee of experts through a Delphi methodology, a GMI intervention based on existing scientific evidence on the implementation of each of its techniques in people with PLP.

## 2. Materials and Methods

### 2.1. Design

The Delphi method is a structured, interactive, and repetitive method that seeks professional consensus among a group of experts [23]. It is applicable when there is limited evidence on a specific topic of interest, allowing recommendations to be derived from the collective judgments of experts [24]. This study consisted of three phases: (1) formation of the committee of experts, (2) online survey, and (3) analysis of the results and elaboration of the conclusions through a final report by the research group. In the first contact with the experts, a questionnaire was sent to carry out a self-assessment on their knowledge and experience in the field of study that allowed us to assess the expertise of each of the possible members. The study was conducted using email and google survey tools as the main methods of communication between the research group and the panel of experts. It consisted in answering a questionnaire about the adequacy of the GMI intervention in patients with PLP. As many rounds as necessary were carried out until consensus was reached among experts the first-round questionnaire was elaborated from a draft of the protocol that was obtained from a systematic review carried out previously. The second-round questionnaire was modified based on the experts’ comments from the first round. It is important to highlight the anonymity of the members of the expert group during the study to avoid possible biases related to the influence of the opinion of the other members [25]. The study flow and objectives for each round are detailed in Figure 1.

### 2.2. Characteristics of the Research Group

The research group was made up of four researchers, three physiotherapists, and an occupational therapist, who are part of the university’s research teaching staff in the areas of health. The number of people in the research group was based on the study by Varela-Ruiz [25], which determines that the research group must be made up of two to five people and its main functions are: selection and recruitment of experts; preparing questionnaires; analyzing the answers of the rounds; answering questions or clarifications by experts; preparing the subsequent questionnaires; interpreting and analyzing the results; and finally writing and submitting the final report.

### 2.3. Intervention Design

The protocol intervention was elaborated through the results obtained from a previous systematic review, where different studies on the effectiveness of graded motor imagery and its components in phantom limb pain in patients with amputations were analyzed.

### 2.4. Ethical Considerations

This study meets the criteria required in the Helsinki Declaration, as well as the Organic Law 3/2018 (5 December) on the Protection of Personal Data and Guarantee of Digital Rights. During the first contact with the participants, informed consent was requested and the information sheet was provided with the contact details of the principal investigator to answer any questions or clarifications if necessary. At the same time, this study was passed by the ethics committee of the University of Vic-Central University of Catalonia with code 143/2021.

### 2.5. Sample Determination

Currently, there are no guidelines or recommendations on the appropriate sample size for expert-consensus Delphi studies, nor is there a standardized definition of a small or large sample size [10]. However, it is established that a minimum sample size of 10 experts who are representative of at least three disciplines is adequate for content validity [26]. Fifteen experts from four different specialties were selected to generate a comprehensive view on the topic.

### 2.6. Participants

Currently, there are no standardized criteria for defining an “expert” in Delphi studies [24]. However, physiotherapists and/or occupational therapists were recruited as evidence shows that it is these two profiles of professionals who use mirror neuron-based techniques in this patient profile [20,21,22]. Professionals from different geographical areas of Spain and South America were included, who completed the different questionnaires, which were all in the common language they shared, Spanish, so that there were no different interpretations due to the language. The inclusion criteria to be part of the group of experts were be specialized in the field of neurorehabilitation and/or pain, with experience in techniques based on mirror neurons, and with a minimum clinical experience of five years in the field of neurorehabilitation and/or pain. The experts were identified through professionals working in areas of amputees or neurology centers where mirror neuron-based techniques were used, and research experience was also taken into account. The experts were contacted via e-mail, where they were asked to participate in the study and provided with the information sheet with all the details of the study and what their task would be as an expert. In addition to the purposive sampling technique, we used snowball sampling, where experts initially recruited by the researcher were encouraged to invite other experts meeting the study’s inclusion criteria from their professional circles to participate in this study [27]. The snowball sampling technique is appropriate for finding additional experts who might not be known by the researcher [28].

In order to ensure the expertise of the participants, the coefficient of expertise (K) of each of them in the field was assessed through a questionnaire via Google forms. This coefficient is defined as the average of the knowledge coefficient (K_c_) and the argumentation coefficient (K_a_). It allowed us to identify the experts, taking into account different characteristics [29]. These characteristics are of an academic, professional, and personal nature and help to classify an individual as an expert in a given subject. The calculation of K is done through four questions that determine the value of K_c_ and K_a_. In the first question, they had to rate from 0 to 10 their level of knowledge in the subjects that intervene in the study (K_c_). In the following three, they had to rate as low, medium, or high, the influence of aspects, such as experience obtained in clinical practice, knowledge of national and international scientific evidence on the subject, and knowledge of technological tools available in the field. These questions constitute the K_a_ value (Table 1). Cabero and Barroso reference values were used for the calculations of the K coefficient [30]:(1)$$\frac{K_{a}+K_{c}}{2}$$

### 2.7. Survey Process

Once the intervention was prepared, four health professionals external to the committee of experts were included to determine the understanding of the questions asked in the questionnaire, as indicated in Varela-Ruiz’s study [25].

After the four health professionals determined that the questions were understood correctly, the questionnaire was sent to the committee of experts. The intervention consisted of three different phases: (1) laterality recognition, (2) motor imagery, and (3) mirror therapy. Prior to each of the three phases, an educational session of approximately 30 min was scheduled, in which the purpose of the phase, the procedure, and the objectives to be achieved were explained and any doubts resolved. Each of the phases contained the explanation in written form and with images to help their understanding and make it more visual. In addition, each of the phases consisted of four questions asking for an assessment of the intensity, frequency, duration, and progression of each of them. The assessment was performed on a 7-point Likert scale, with 1 strongly disagreeing and 7 strongly agreeing. This scale is efficient and easy to use and has been validated to facilitate the experts’ rating of items in Delphi studies [31]. Finally, there was an open question at the end of each phase so that each expert could make the necessary contributions. The time given for the experts to answer the questionnaire was two weeks, with a reminder email being sent in the last few days.

Once the first-round responses were received, they were statistically analyzed, and the second-round questionnaire was based on the results of the first round. In the second round, the results obtained from each question in the first round were shown, and the questionnaire modified according to the comments received. The questionnaire of this second round specified those aspects that were modified with respect to the first based on the suggestions and comments received by the experts in the open-ended questions of each phase [24]. Comments and suggestions had to be substantiated with scientific evidence so that the research group could assess and verify the modification. In this second round, the questions were asked again using the same 7-point Likert scale so that the expert could score again based on the changes that had been applied [32]. As many rounds as necessary were conducted in order to reach consensus among the experts.

### 2.8. Statistical Analysis

At the end of the first round, the statistical analysis was carried out. In each of the rounds, through the Excel and R commander 3.0.2 programs, we analyzed the mean, median (Me), maximum, minimum, standard deviation (SD), first quartile (Q1), third quartile (Q3), and interquartile range (IQR = Q3−Q1) and relative interquartile range (RIR = IQR × 100/Me) of each of the questions related to each phase.

To determine the validation of each of the aspects, the fact that the interquartile range was as close to 0 as possible, and the RIR was less than 15%, were both taken into account [19,20].

In parallel with each of the aspects to be assessed, the Kappa index was calculated at the end of the second round. It is based on comparing the concordance observed in a data set, with respect to what could be given due to chance [2] (see Table 2). This index was only calculated in the final phase when consensus between experts was reached.

## 3. Results

### 3.1. Expert Panel

A total of 15 participants were part of the committee of experts. In total, 92.9% of the people who formed the committee were physiotherapists and 7.1% were occupational therapists, with an average of 11.2 years of clinical experience. The median age was 31.5 years, with a minimum of 27 years and a maximum of 42 years. In total, 64.3% of the participants were experts in neurorehabilitation and the remaining 35.7% in pain.

The assessment of the coefficient of expert showed an average of 0.78 (K_c_), 0.89 (K_a_), and 0.83 (K). The level of expertise of the committee of experts is considered high [13]. The different values are shown in Table 3 below.

In the first round, a consensus was reached in seven of the assessed aspects—the duration and progression of the three techniques and the frequency established in mirror therapy—as they presented an IQR close to 0 and an RIR less than 15%. They can thus be said to have been validated by the experts in the first round.

The changes made in the second round were to increase the frequency of the first two techniques (laterality recognition and motor imagery) to twice a day, thus also increasing the intensity of the intervention during the day. Finally, the intensity of the mirror therapy intervention was increased to 20 min per day.

In the second round, 13 of the 15 participants responded to the protocol, the same number that was achieved in the first round. During this phase, a consensus was reached on the entire intervention by the experts.

### 3.2. Validation

Table 4, Table 5, Table 6 below show the results obtained from the statistical analysis in each round of the three techniques.

In the second round, the Kappa index was calculated to observe the strength concordance with respect to chance. It was 0.76, which indicates a good concordance [2].

Figure 2 below shows the sequence and progression of the intervention. The validated protocol is called GraMI. The GraMI protocol consists of performing three progressive techniques over time, with an educational session prior to the start of each technique to explain the procedure and objectives and to resolve any possible doubts and encourage monitoring of the intervention. Each technique progresses in difficulty over time. This protocol is easy to implement, allowing the patient to perform it autonomously and individually at home with the follow-up of a professional. The GraMI protocol is detailed in the Appendix A.

## 4. Discussion

Given the lack of protocols on GMI in PLP, this study used a Delphi methodology to seek consensus among experts and to develop and validate a GMI intervention in patients with amputations that experience PLP (GraMI protocol). The systematic review by Herrador Colmenero (2017) suggests that new treatment protocols should specify the type of intervention and the frequency, duration, and progression of the sessions [33].

GMI is a set of three techniques of progressive difficulty that the participant can perform autonomously under professional supervision and be active throughout the process. The success of treatment can often depend on the degree of patient participation: the greater the participation, the greater the benefits obtained [34]. GMI seeks the reorganization of brain plasticity through laterality recognition, motor imagery, motor work, sensory interaction with objects, and functional activities through observation and movements of the ipsilateral side to areas injured by amputation. Observation is one of the techniques based on mirror neurons, where it is established that the same brain activation occurs whether the movement is done in the first person or is observed in a third person or, in the case of mirror therapy, in the reflection of the mirror [14].

During the first round, the experts suggested increasing the intensity and frequency of interventions to twice a day compared to just once, as suggested by some authors [19,22,35]. This increase in frequency allows for a more intensive intervention. After an amputation, there is poor plasticity and a more intensive and lasting intervention over time can again lead to favorable neuroplastic changes [36,37].

Within each of the stages is a progression in difficulty of the images ending in interaction with objects and functional activities. This progression aims to enhance plasticity and not allow it to stagnate, thus encouraging motor learning [38]. In the first two stages, laterality recognition and motor imagery, we begin by identifying each of the images taken in neutral positions for five consecutive days. Next, different images taken in different planes over an equal period of five more days are identified. Finally, in order to achieve motor learning of what is being worked on, it is important to transfer the movement that is being identified to daily life, that is, in functional activities and interaction with objects, since if this transfer is not made, it is not considered that a motor learning has been achieved [39].

In the third stage, mirror therapy, consensus was reached among experts to progress from analytical movements of the joints involved in amputation to sensory stimulation and, finally, to an interaction with functional objects and activities. This progression has been proposed and validated with the idea of working all the components that form an action, from the motor and sensory part necessary to plan and organize the movement through the pyramidal path, to the interaction with objects and functional activities to make it applicable to daily life and promote motor learning for each action [38]. In addition, the movement of the healthy limb, in the phase of mirror therapy, produces an activation of the ipsilateral fibbers of the pyramidal pathway that do not cross at the level of the bulb, producing a preactivation of the injured areas [40].

### Limitations and Strengths

This study is the first validated GMI intervention by experts in neurorehabilitation and pain. Having perspectives from different groups provides us with a more comprehensive vision. The questionnaire is written in Spanish, which is one of the experts’ main languages. As a result, the language, and consequently the translation of the questionnaire, does not influence its interpretation. Finally, the use of a seven-point Likert scale represents a strength, as it shows better internal consistency in this type of study than the five-point Likert scales.

One limitation of the study is the lack of existing scientific evidence on GMI in PLP. The design of the intervention is based on studies that used some of the techniques that make up GMI, as there are very few studies that use a combination of the three techniques.

## 5. Conclusions

The GraMI protocol was validated by a group of experts, presenting a good level of acceptance and agreement using the Delphi method. It could be expected that this intervention, implemented intensively for 15–30 min a day for nine weeks, can decrease PLP in patients with amputations. In order to study the effectiveness of the GraMI protocol, the next step will be to perform a randomized clinical trial.

## Figures and Tables

**Figure 1 ijerph-18-12240-f001:**
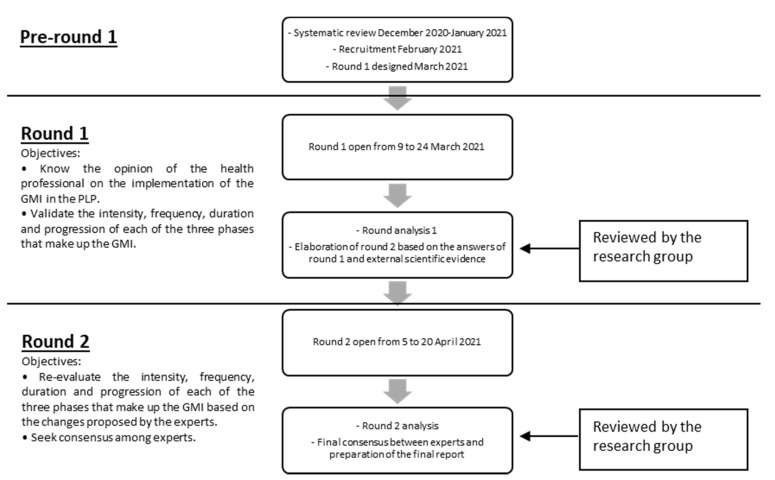
Study flow and objectives.

**Figure 2 ijerph-18-12240-f002:**
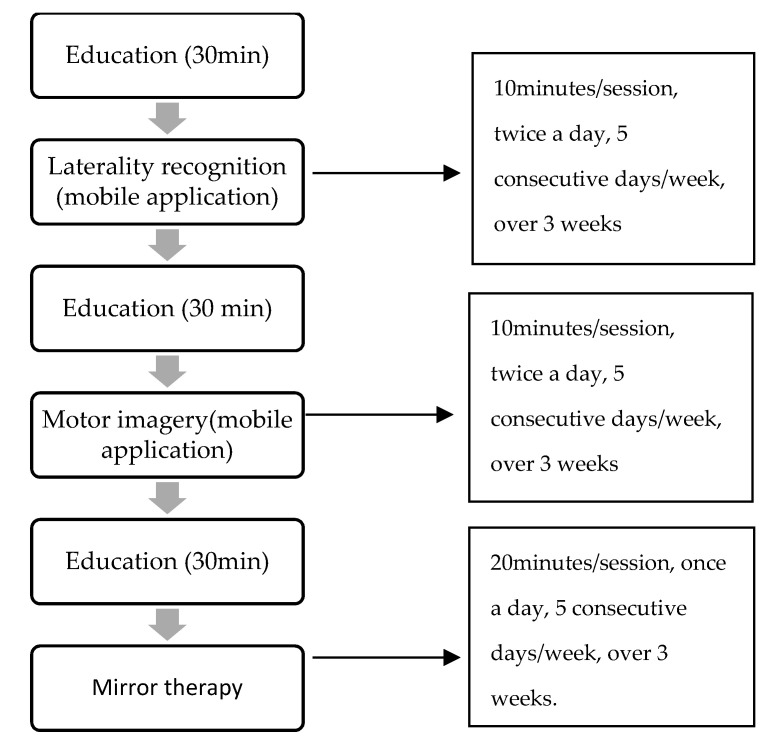
Sequence and progression of the intervention.

**Table 1 ijerph-18-12240-t001:** Values used to assess the argumentation coefficient (Ka) [29].

	High	Medium	Low
Theoretical analysis performed by the expert	0.3	0.2	0.1
Experience gained	0.5	0.4	0.2
Studies on the subject by Spanish authors	0.05	0.05	0.05
Studies on the subject by international authors	0.05	0.05	0.05
Own knowledge	0.05	0.05	0.05
Expert intuition	0.05	0.05	0.05

**Table 2 ijerph-18-12240-t002:** Kappa index rating (K).

Value of K	Strength of Concordance
<0.20	Poor
0.21–0.40	Weak
0.41–0.60	Moderate
0.61–0.80	Good
0.81–1.00	Very good

**Table 3 ijerph-18-12240-t003:** Values obtained in each of the participants.

Participant	K_c_	K_a_	K
1	0.7	0.9	0.80 medium
2	0.8	0.9	0.85 high
3	1	0.9	0.95 high
4	0.8	1	0.90 high
5	0.7	0.9	0.8 medium
6	0.7	0.9	0.8 medium
7	0.7	0.9	0.8 medium
8	0.9	0.8	0.85 high
9	0.8	0.8	0.8 medium
10	0.7	0.9	0.8 medium
11	0.7	0.8	0.75 medium
12	0.7	0.8	0.75 medium
13	0.7	0.9	0.80 medium
14	0.9	1	0.95 high
15	0.7	0.9	0.80 medium
Mean	0.77	0.89	0.83 high

K_c_: knowledge coefficient K_a_: argumentation coefficient K: coefficient of expert competence.

**Table 4 ijerph-18-12240-t004:** Laterality recognition.

	R1 Intensity	R2 Intensity	R1 Frequency	R2 Frequency	R1 Duration	R2 Duration	R1 Progression	R2 Progression
Mean	6.16	6.53	5.84	6.46	6.5	-	6.69	-
Median	6	7	7	7	7	-	7	-
Maximum	7	7	7	7	7	-	7	-
Minimum	4	6	2	3	4	-	6	-
Standard deviation	0.89	0.52	1.72	1.09	0.87	-	0.48	-
Quartile 1	6	6	6	6	6	-	6	-
Quartile 3	7	7	7	7	7	-	7	-
IQR	1	1	1	1	1	-	1	-
RIR (%)	16.6%	14.28%	14.28%	14.28%	14.28%	-	14.28%	-

R1: Round 1; R2: Round 2; IQR: Interquartile range; RIR: Relative interquartile range; -: Validated in the first round.

**Table 5 ijerph-18-12240-t005:** Motor imagery.

	R1 Intensity	R2 Intensity	R1 Frequency	R2 Frequency	R1 Duration	R2 Duration	R1 Progression	R2 Progression
Mean	6.07	6.77	6.07	6.54	6.7	-	6.7	-
Median	6	7	6	7	7	-	7	-
Maximum	7	7	7	7	7	-	7	-
Minimum	3	6	3	4	6	-	6	-
Standard deviation	1.18	0.44	1.18	0.88	0.63	-	0.63	-
Quartile 1	6	7	6	7	7	-	7	-
Quartile 3	7	7	7	7	7	-	7	-
IQR	1	0	1	1	0	-	0	-
RIR (%)	16.6%	0%	16.6%	14.28%	0%	-	0%	-

R1: Round 1; R2: Round 2; IQR: Interquartile range; RIR: Relative interquartile range; -: Validated in the first round.

**Table 6 ijerph-18-12240-t006:** Mirror therapy.

	R1 Intensity	R2 Intensity	R1 Frequency	R2 Frequency	R1 Duration	R2 Duration	R1 Progression	R2 Progression
Mean	6.15	6.77	6.4	-	6.7	-	6.7	-
Median	7	7	7	-	7	-	7	-
Maximum	7	7	7	-	7	-	7	-
Minimum	3	4	4	-	6	-	5	-
Standard deviation	1.34	0.44	0.96	-	0.48	-	0.64	-
Quartile 1	6	7	6	-	6	-	7	-
Quartile 3	7	7	7	-	7	-	7	-
IQR	1	1	1	-	1	-	0	-
RIR (%)	14.28%	14.28%	14.28%	-	14.28%	-	0%	-

R1: Round 1; R2: Round 2; IQR: Interquartile range; RIR: Relative interquartile range; -: Validated in the first round.

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
