# Peer review of "Development and Validation of a Graded Motor Imagery Intervention for Phantom Limb Pain in Patients with Amputations (GraMI Protocol): A Delphi Study"

_ijerph, 2021, doi:10.3390/ijerph182212240_

Round 1

Reviewer 1 Report

Dear editor, dear authors,

Please find below my review of the manuscript “Development and Validation of a Graded Motor Imagery Intervention for Phantom Limb Pain in Patients with Amputations (Grami Protocol): A Delphi Study” on how to apply Motor Imagery Therapy

I feel that the authors really put a lot of time and effort into this Delphi study and the subsequent manuscript. Overall, the authors did great work but could not properly convey all aspects of it, due to language issues in the manuscript and conceptualisation. Some facts seem to be lost in translation and sometimes the language is not scientifically appropriate. It would be a shame if all the work that went into this study would not be fully addressed in the manuscript. For example, it is not addressed, that not all patients are even capable of imagining a limb, where theirs is lost, what should be done in this case? Are there exercises to do with these patients before starting the protocol? Also, at passages, the text is rather repetitive and there are errors that appear to have occurred due to inattentiveness, such as citations that are not in brackets.

Please find my detailed review below:

Abstract: PLP is not restricted to surgical removal of the limb

repetitions (pain)

the series of rounds itself were not established, but the protocol

L30: again, not limited to surgical removal

L34: What does time of evolution mean?

L51: Surgical treatment as in TMR? This should be mentioned explicitly, since only surgical removal was discussed, reader might think of further amputation

L75: if CRPS is mentioned, it should be explained how it is different from PLP

L76: citations

L 78: citations

L83: language

L90: do the authors really mean “telematic” or maybe “online”?

There should be more references mentioned in the introduction, there is a bit more to say about motor imagery. It would be even better, if the authors would do a scoping review to bas their Delphi study on scientific evidence.

If the goal is to create a protocol that all therapists should use, it is important to show how treatment has been done so far.

L97: was, not were

Repetition

L107: Physicians? Therapists ? What kind of doctors? Please specify

L113f: language

In which

L145f: language

L213: sentence needs rewriting

L 234: the grammar in this sentence is off, also, the authors state that the protocol is explained in greater detail in the annex – where is the annex? It is not in the supplementary material section.

Figure 2: What is meant by education? For how long should this treatment be followed? In L291 the authors state that the intervention should last 6 weeks, so it should be made clear in the flowchart, that the mentioned interventions are not done in parallel.  Also, a bit more information on this main outcome of the delphi study would be beneficial.

Tables: Some tables could be graphs instead, that are easier to understand by just looking at them.

All abbreviations used in the tables and figures should be explained directly within the table or in the subtext.

Language needs to be heavily revised, there are major understanding problems in this manuscript.

References: please add when the homepages were accessed

Reviewer 2 Report

Using Delphi method (question and survey based method among experts to find out  a better solution for a problem), authors studied how graded motor imagery, a physiotherapeutic treatment, for phantom limb pain can be improved. At the first round they found experts on this field and then at the second round they questioned quality and quantification of GMI treatment. There was a good agreement among experts regarding methodology. Authors suggest how many sessions and how long each session are needed for a better outcome for physiotherapeutic approach for treatment of phantom limb pain. It would be good for readers to know what the next step for authors. Are they planning a clinical study? Will they use the suggested methods for their patients? These answers can be added to the conclusion section. Additionally, very minor correction is there are references in the text not placed in parenthesis.
